# Prediction of Survival by IL-6 in a Randomized Placebo-Controlled Trial of Anakinra in COVID-19 Cytokine Storm

**DOI:** 10.3390/v15102036

**Published:** 2023-09-30

**Authors:** Lesley E. Jackson, Nitasha Khullar, Timothy Beukelman, Chris Chapleau, Abhishek Kamath, Randy Q. Cron, Walter Winn Chatham

**Affiliations:** 1Division of Clinical Immunology and Rheumatology, University of Alabama at Birmingham, Birmingham, AL 35233, USAwchatham@uabmc.edu (W.W.C.); 2Children’s of Alabama, Birmingham, AL 35233, USA; 3UAB Hospital Pharmacy, University of Alabama at Birmingham, Birmingham, AL 35233, USA; 4Heersink School of Medicine, University of Alabama at Birmingham, Birmingham, AL 35233, USA

**Keywords:** SARS-CoV-2, antiviral treatment, anakinra

## Abstract

(1) Background: Some severe COVID-19 patients develop hyperinflammatory cytokine storm syndrome (CSS). We assessed the efficacy of anakinra added to standard of care (SoC) in hospitalized COVID-19 CSS patients. (2) Methods: In this single-center, randomized, double-blind, placebo-controlled trial (NCT04362111), we recruited adult hospitalized patients with SARS-CoV-2 infection, evidence of pneumonia, new/increasing oxygen requirement, ferritin ≥ 700 ng/mL, and at least three of the following indicators: D-dimer ≥ 500 ng/mL, platelet count < 130,000/mm^3^, WBC < 3500/mm^3^ or lymphocyte count < 1000/mm^3^, AST or ALT > 2X the upper limit of normal (ULN), LDH > 2X ULN, C-reactive protein > 100 mg/L. Patients were randomized (1:1) to SoC plus anakinra (100 mg subcutaneously every 6 h for 10 days) or placebo. All received dexamethasone. The primary outcome was survival and hospital discharge without need for intubation/mechanical ventilation. The data were analyzed according to the modified intention-to-treat approach. (3) Results: Between August 2020 and January 2021, 32 patients were recruited, of which 15 were assigned to the anakinra group, and 17 to the placebo group. Two patients receiving the placebo withdrew within 48 h and were excluded. The mean age was 63 years (SD 10.3), 20 (67%) patients were men, and 20 (67%) were White. At Day 10, one (7%) patient receiving anakinra and two (13%) patients receiving the placebo had died (*p* = 1.0). At hospital discharge, four (27%) patients receiving anakinra and four (27%) patients receiving the placebo had died. The IL-6 level at enrollment was predictive of death (*p* < 0.01); anakinra use was associated with decreases in CXCL9 levels. (4) Conclusions: Anakinra added to dexamethasone did not significantly impact the survival of COVID-19 pneumonia patients with CSS. Additional studies are needed to assess patient selection and the efficacy, timing, and duration of anakinra treatment for COVID-19 CSS.

## 1. Introduction

The respiratory illness coronavirus disease 2019 (COVID-19) caused by severe acute respiratory syndrome coronavirus 2 (SARS-CoV-2) induces a moderate to severe disease requiring hospital admission and supplemental oxygen support in approximately 10–15% of patients [1,2,3]. A subset of this population will develop the life-threatening hyperinflammatory response consistent with cytokine storm syndrome (CSS) [4,5]. Anakinra is a recombinant interleukin-1 receptor antagonist used to treat some inflammatory disorders [6] and previously demonstrated mortality benefit in sepsis patients with CSS [7]. Anakinra may reduce hyperinflammation and improve outcomes in some patients with COVID-19-associated CSS [8,9].

Thus far, data on the efficacy of anakinra in patients with COVID-19-associated CSS have been largely driven by observational studies [10,11,12,13,14,15,16,17] or non-randomized open-label clinical trials [18,19,20]. The first reported randomized control trial was terminated early for concerns that anakinra was not improving outcomes in patients with mild-to-moderate COVID-19 pneumonia [21]. Two meta-analyses suggested a COVID-19 survival benefit of anakinra, also among those patients with signs of hyperinflammation [22] and in the absence of glucocorticoid co-administration [22,23]. 

The objective of this investigator-initiated randomized, double-blind, placebo-controlled trial was to assess the efficacy and safety of anakinra when added to usual care for the treatment of CSS in patients with COVID-19 pneumonia to prevent disease progression and complications. We hypothesized that early intervention with interleukin-1 receptor blockade might limit the progression to hypoxemic respiratory failure, requiring intubation and mechanical ventilation, or death, reduce the risk of clinical worsening, improve markers of inflammation, and decrease the supplemental oxygen requirement. 

## 2. Materials and Methods

### 2.1. Study Design and Participants

This double-blind, randomized, placebo-controlled trial was conducted at a large, urban academic medical center. The trial was approved by the University of Alabama at Birmingham (UAB) Institutional Review Board and was conducted in accordance with Good Clinical Practice guidelines. All patients provided written informed consent consistent with the institutional guidelines. The investigators designed the clinical trial, collected all the data, and performed the analysis. The UABSOM COVID-19 Research Initiative and Swedish Orphan Biovitrum (SOBI) funded the trial and provided anakinra and placebo, respectively, but had no role in data analysis, data interpretation, or writing of the original manuscript. The trial was registered with ClinicalTrials.gov, NCT04362111. 

Patients were eligible for enrollment if they had nucleic acid- (PCR RNA) confirmed SARS-CoV-2 infection, mild-to-moderate COVID-19 pneumonia based on imaging, new or increasing oxygen requirement to maintain oxygen saturation greater than 93%, and serum ferritin level greater than 700 ng/mL. Patients had to have at least three of the following additional criteria: elevated D-dimer more than 500 ng/mL, platelet count less than 130,000/mm^3^, white blood cell count less than 3500/mm^3^ or lymphocyte count less than 1000/mm^3^, elevated AST or ALT greater than two times the ULN, elevated lactate dehydrogenase (LDH) greater than two times the ULN, C-reactive protein (CRP) greater than 100 mg/L. Patients on mechanical ventilation or with a confirmed active bacterial infection requiring antibiotic therapy were excluded. Patients were excluded if there was concern for a developing infection or if the investigator believed adding anakinra would place them at higher risk for an infection. The use of immunosuppressants such as tocilizumab or sarilumab was not standard of care at the time this trial was conducted very early in the COVID-19 pandemic, and we excluded patients receiving any other immunomodulatory agents. The full list of entry criteria is provided in the ClinicalTrials.gov protocol. 

### 2.2. Randomization and Masking

Eligible patients were randomly assigned (1:1) using a web-based secure centralized system to either standard care plus anakinra (100 mg subcutaneously every 6 h for 10 days) or placebo (normal saline subcutaneously every 6 h for 10 days) plus standard care. The administration of anakinra or placebo was generally initiated within 12 h after informed consent was obtained. All patients received dexamethasone (6 mg daily) as concomitant treatment based on the results of the Randomized Evaluation of COVID-19 Therapy (RECOVERY) trial [24]. The dose frequency of anakinra was decreased to twice daily after Day 5 if the levels of ferritin, D-dimer, CRP, LDH, AST, ALT had all returned to normal or improved by a value greater than or equal to 75% with respect to the noted Day 0 elevations and the subject was maintaining oxygen saturation of at least 93% on room air. 

### 2.3. Data Collection and Outcomes

The primary outcome was survival and discharge from the hospital without the need for intubation/mechanical ventilation. The secondary outcomes included (1) no increase in oxygen requirement or oxygen delivery measures to maintain oxygen saturation > 90% from Day 2 (48 h) to Day 10 (120 h); (2) at least a 25% decrease in serum ferritin, LDH, CRP, and D-dimer at 48 h; (3) normalization or ≥ 75% improvement by Day 10 (120 h) in each of the following laboratory CSS attributes if elevated beyond the normal range at randomization: ferritin, fibrinogen, AST, ALT, leucopenia, thrombocytopenia, D-dimer, CRP, triglycerides; (4) decreased time from the initial dosing of anakinra or placebo to the achievement of ≥ 93% oxygen saturation on room air for 24 h; and (5) no increased prevalence of nosocomial bacterial, fungal, or non-COVID-19-related viral infection through the time of hospital discharge until Day 28. As an exploratory outcome, we also collected blood for the assessment of a panel of cytokines that included interleukin-2 (IL-2), soluble IL-2 receptor-alpha (sCD25), IL-12, IL-4, IL-5, IL-10, IL-13, IL-17, IL-6, IL-8, interferon gamma (IFN-γ), IL-1β, tumor necrosis factor, CXCL9, and sCD163 (soluble haptoglobin receptor), collected at Day 0 and Day 10. IL-18 levels were assayed at Day 0.

Until the date of hospital discharge (prior to or subsequent to Day 10), the occurrence of the following events was recorded according to a time-to-event analysis: requirement for mechanical ventilation, withdrawal of oxygen therapy due to lack of requirement to maintain RA oxygen saturation ≥ 93%, and/or death of the study subject. 

The patients were reassessed at Day 28 and again at Day 60 and by electronic health record review at Day 90. At both Day 28 and Day 60, any new viral, bacterial, or fungal infections were recorded. The current oxygen support or the last day of needed oxygen support subsequent to hospital discharge, and any need for re-hospitalization were recorded. The data from patients who were event-free at the end of the follow-up were censored at 90 days for the primary and secondary outcomes. 

### 2.4. Statistical Analysis

We assumed the subset of admitted patients with CSS attributes would have a very high mortality (80%) and, wishing to determine if anakinra use added to standard of care would be significantly impactful, we determined that a sample size of 32 participants (16 in each group) would provide an adequate power of 80% using a two-side alpha of 0.05 to detect a clinically relevant reduction in mortality of 60% between the two groups. The primary efficacy analysis was conducted on a modified intention-to-treat basis and was defined as all patients who underwent randomization and received either anakinra or placebo before intubation/mechanical ventilation or death. Descriptive statistics were used to characterize the participants. Continuous data were summarized by means and standard deviation (SD) or standard error of the mean (SEM, for the cytokine data) and analyzed by Student’s *t* tests. Categorical data were described with frequencies and percentages and compared using Fisher’s exact or chi square tests. A blinded interim analysis was conducted by an independent safety monitoring committee after the enrollment of 15 patients. Cox proportional hazards models were used to calculate the hazard ratio for the association between treatment group and death from any cause or need for mechanical ventilation; due to the small sample size, controlling for demographic characteristics, clinical variables, and comorbidities was not possible. We calculated the 95% confidence interval (CI), and *p* < 0.05 was considered significant. All analyses were conducted with SAS v9.4 (Cary, NC, USA). 

## 3. Results

### 3.1. Patients

Between 5 August 2020 and 2 January 2021, 235 patients were prescreened for meeting the entry criteria, and 32 patients were recruited for participation in the study; of them, 15 were assigned to the anakinra plus standard care group, and 17 to the placebo plus standard care group (Figure 1). Two patients in the placebo group withdrew in the initial 48 h and were excluded from the modified intention-to-treat analysis. The baseline characteristics of the participants are displayed in Table 1 and Table 2. The treatment groups were balanced. In the analyzable population, the mean age was 63 years (SD 10.3), 20 (67%) participants were men, 20 (67%) participants were White, and 9 (30%) were Black or African American. 

During the treatment duration, all participants received dexamethasone (6 mg daily) as adjunctive treatment by the primary managing service. Similar percentages of patients in the two groups received remdesivir, antibiotics, and anticoagulation. Remdesivir was administered to 12 patients (80%) in the anakinra group and 14 patients (93%) in the placebo group. Azithromycin was administered to 10 patients (67%) in the anakinra group and 8 patients (53%) in the placebo group. β-lactam antibiotics were administered to 13 patients (87%) in the anakinra group and 10 patients (67%) in the placebo group. Prophylactic anticoagulation was administered to 12 patients (80%) in the anakinra group and 15 patients (100%) in the placebo group.

### 3.2. Primary Outcome

For the primary outcome of survival and hospital discharge without need for intubation/mechanical ventilation, there were 10 (66.7%) patients in the anakinra group and 10 (66.7%) patients in the placebo group that met this endpoint (*p* = 1.0) (Table 3). At Day 10, one (7%) patient in the anakinra group and two (13%) patients in the placebo group had died (*p* = 1.0) (Table 3). At Day 10, four (27%) patients in the anakinra group and three (20%) patients in the placebo group had required invasive mechanical ventilation (*p* = 1.0). At hospital discharge, four (26.7%) patients in the anakinra group and four (26.7%) patients in the placebo group had died (*p* = 1.0). 

The Kaplan–Meier curve for death from any cause is shown in Figure 2. The unadjusted hazard ratio for the outcome event of death from any cause in the anakinra group was 0.94 (95% CI: 0.24–3.8). Due to the small number of outcomes, there was no adjustment for age, sex, or co-morbidities. As illustrated in the Kaplan–Meier curve, two deaths in the anakinra arm and all four deaths in the placebo arm occurred before Day 30. 

### 3.3. Secondary Outcomes and Exploratory Outcomes

The secondary outcomes are displayed in Table 3. After Day 2, eight (53%) patients in the anakinra group and nine (60%) patients in the placebo group did not require escalation of oxygen support (*p* = 0.7). At Day 2, six (40%) patients in the anakinra group and eight (57%) patients in the placebo group demonstrated 25% improvement in Day 0 elevations of serum CRP and ferritin (*p* = 0.4). At Day 10, six (46%) patients in the anakinra group and seven (54%) patients in the placebo group demonstrated 75% improvement in Day 0 elevations of serum CRP and ferritin (*p* = 0.7). At Day 10, no (0%) patients in the anakinra group and seven (54%) patients in the placebo group demonstrated 75% improvement in Day 0 LDH and D-dimer elevations (*p* = 0.005). At Day 10, eight (57%) patients in the anakinra group and nine (69%) patients in the placebo group were requiring no more than 2 L/min of nasal cannula oxygen support to maintain an oxygen saturation ≥93% (*p* = 0.7). At Day 10, four (29%) patients in the anakinra group and nine (69%) patients in the placebo group were able to maintain an oxygen saturation ≥93% on room air (*p* = 0.06). 

There were no statistically significant differences in the mean CRP concentration, mean ferritin concentration, mean lactate dehydrogenase (LDH) concentration, and mean D-dimer concentration over time between the patients in the anakinra and placebo care groups at enrollment (Day 0), at Day 2, or at Day 10 post enrollment (data not shown). Confirmed microbial infections occurred in four (27%) patients in the anakinra group (of which one was fungemia caused by candida species, and the remaining infections were considered colonized organisms) and two (13%) patients in the usual care group, which were both microbial colonizations (*p* = 1.0). No patients in the anakinra group demonstrated improvement of coagulopathy markers (including LDH and D-dimer) by Day 10, but there was also no association of the use of anakinra with the development of venous thromboembolism during the hospitalization. There were no statistically significant associations of race/ethnicity, presence of underlying co-morbid diabetes, hypertension, or malignancy with survival or need for mechanical ventilation. 

Regarding the observed cytokine levels, only the levels of IL-6, IL-10, IL-18, and CXCL9 were noted to be significantly elevated (i.e., detectable and above the reference range for normal in over 50% of the subjects) at the time of enrollment (see Table 4). The placebo and anakinra groups were evenly balanced, with the exception of significantly higher levels of CXCL9 in the anakinra group (mean 200 ± SEM 42 for the placebo group vs. 466 ± 100 for the anakinra group, *p* = 0.03). The mean CXCL9 levels did not significantly change from Day 0 to Day 10 in the placebo group but decreased in the anakinra group (Day 0, 466 ± 100, Day 10, 169 ± 30 ng/mL, *p* = 0.01, Welch’s *t*-test for groups with unequal variance). Restricting the analysis to just those subjects for whom paired Day 0 and Day 10 data were available, the mean decrease in CXCL9 levels was greater in the anakinra group but did not reach statistical significance (*t*-test for paired data). The only cytokine level at enrollment predictive of death was that of IL-6 (survivors, mean 4.3 ± SEM 0.7 ng/mL vs. deceased, 21.9 ± SEM 11.9 ng/mL, *p* < 0.02). 

### 3.4. Safety

The recorded adverse events are displayed in Table 5. No new safety signals for anakinra were identified. Of the four patients in the anakinra group who died, three progressed to respiratory failure: one of them required extracorporeal membrane oxygenation (ECMO) and expired on Day 79, the second was intubated on Day 0 after having received just one dose of anakinra and later expired on Day 6, and the third developed progressive respiratory failure requiring intubation on Day 8 and later expired on Day 13. The fourth patient developed an acute abdomen on Day 2 following anakinra initiation, with CT angiography demonstrating extensive thrombotic disease of the aorta with superior mesenteric artery occlusion and ischemic bowel on laparotomy and ultimately succumbed to complications of ischemic bowel on Day 33. 

Of the four patients in the placebo arm who died, three progressed to respiratory failure: the first developed progressive respiratory failure and expired on Day 7, the second demonstrated initial respiratory status improvement by Day 6 but, following dexamethasone discontinuation, clinically deteriorated and expired on Day 16, and the third rapidly progressed to respiratory failure requiring intubation on Day 3 with ECMO by Day 4 and ultimately expired on Day 27. The fourth death in the placebo arm occurred on Day 2, and the autopsy showed significant pulmonary arteriolar thrombi. 

## 4. Discussion

The primary outcome clinical trial data of this study do not provide support for the use of anakinra in the treatment of severe CSS in COVID-19 pneumonia. The hypothesis underlying the study was that interleukin-1 receptor blockade in patients with COVID-19 would disrupt the cytokine storm and prevent the progression of the respiratory decline. The findings of this randomized, double-blind, placebo-controlled trial indicated that anakinra did not demonstrate a significant effect on the need for mechanical ventilation, death, the improvement in markers of inflammation (except for a nonsignificant improvement in CXCL9 levels, see below), or the need for supplemental oxygen. However, we found that a lower initial level of IL-6 predicted survival in this patient population. Given the small sample and the width of the confidence intervals in the efficacy comparisons, the possibility that treatment with anakinra may be associated with some clinical benefit or harm in certain patients cannot be excluded. 

The results reported herein vary from the conclusions drawn by several recent studies which suggested that interleukin-1 receptor blockade may have a positive effect on patients with COVID-19-associated CSS [8]. Previous evidence was largely driven by observational studies or open-label trials with a high risk of bias. Recently, a large randomized, double-blind, placebo-controlled trial reported a survival benefit of early anakinra administration (largely, in addition to glucocorticoids), where patient selection was based on elevated serum levels of soluble urokinase plasminogen activator receptor (suPAR) [25]. This study found that the IL-6 levels decreased in the group treated with anakinra, and similarly, that the initial IL-6 level was predictive of clinical outcomes including survival, as we also found in our study. This trial utilized a lower once-daily dose of anakinra for 7–10 days compared to our study. The inclusion of this suPAR-based clinical trial in an updated meta-analysis demonstrated that early anakinra administration reduced mortality in hospitalized COVID-19 patients, especially among those with higher initial concentrations of CRP [23,25]. Of note, this review concluded that anakinra was associated with a survival benefit primarily when administered without concomitant dexamethasone. All patients in our study received dexamethasone, as this became standard of care during the study duration and thus likely played a role in improving outcomes in our population. Moreover, the authors found that anakinra was particularly effective in decreasing the risk of mortality in people with CRP concentrations higher than 100 mg/L. However, approximately 30% of our population had enrollment CRP concentrations less than 100 mg/L (the level that was associated with a mortality benefit in the systematic review), a finding which may help explain the lack of significant benefit seen in our study [23].

Although not reaching statistical significance in the paired data analysis, the observation in our study that the CXCL9 levels trended lower among those treated with anakinra is noteworthy. CXCL9 is an interferon-gamma response gene and a recognized serum biomarker for increased IFN-γ production [26]. In several disorders associated with hyperinflammation, such as severe systemic lupus erythematosus or HIV/AIDS, elevated levels of suPAR were shown to correlate with increased expression of IFN-γ or IFN-γ response genes [27,28]. The reported improved outcomes in COVID-19 patients with higher levels of inflammatory markers such as suPAR and CRP after treatment with anakinra may therefore reflect favorable impacts of IL-1 blockade on IFN-γ activation and/or IFN-γ effector responses. 

Anakinra dosing and duration vary widely among the published studies. An observational study which demonstrated improvement in clinical outcomes with anakinra utilized substantially higher doses and for a longer duration to suppress inflammation [10] than in the current study. Another observational study utilized lower doses of anakinra and demonstrated an association with a reduction in a composite outcome of invasive mechanical ventilation or death (HR 0.2, CI 0.1–0.5) [11]. Among the prospective open-label trials, some studies utilized anakinra initially either once [19] or twice daily [18] and demonstrated an association with a reduced need for mechanical ventilation [18,19] and an improvement in inflammatory biomarkers [18,19]. In two open-label randomized controlled trials of anakinra in patients with COVID-19-associated CSS, the dose and administration frequency of anakinra were lower than those we used in this current study [21,29]. One study demonstrated a significantly decreased need for mechanical ventilation in the group which received anakinra compared to the control group [29], while the other trial did not demonstrate improvement in clinical outcomes [21]. 

A major strength of this trial is its design as a double-blind randomized study of non-intubated patients with COVID-19-associated CSS. Our entry criteria were strict compared to those of most previous studies on COVID-19, regardless of the treatment explored. Despite the small sample size, the anakinra and placebo groups were well balanced according to baseline characteristics and additional treatments received during the study duration. 

Despite its strengths, this study has a few limitations. The explanation for the failure of anakinra to improve clinical outcomes in this trial is not clear. This study was conducted in a single center, which may limit the generalizability of our findings. This small study might be underpowered to detect statistically significant differences in intervention effects between groups. Appropriate sample sizes could not be accurately estimated when the trial was being planned near the onset of the COVID-19 pandemic. However, at the time of the study conceptualization, the in-hospital mortality was estimated to be approximately 80% among those admitted with CSS features. With the advent of routine dexamethasone use between the time of study conceptualization and submission for regulatory approvals and the start of enrollment, mortality was not as high as originally projected, and the study, as designed, became underpowered. It is possible that certain subgroup populations may nonetheless benefit from interleukin-1 receptor blockade using a different dose or duration with respect to those used in this study. This may be addressed in a larger randomized double-blind study. Lastly, the entry criteria were perhaps the strictest applied in any COVID-19 trial in regard to the requirement for presenting multiple features of CSS. Less stringent entry criteria, or using suPAR levels alone, may reveal the benefit of anakinra in treating hospitalized/hypoxic COVID-19 patients prior to the requirement for invasive mechanical ventilation [30]. 

In summary, anakinra added to dexamethasone did not significantly impact patient outcomes in this study of patients with clinical laboratory features of severe CSS and mild-to-moderate COVID-19 pneumonia. Additional larger studies are needed to assess the efficacy, safety, timing, and optimal dosing and duration of anakinra treatment in select patients with more severe COVID-19 [31]. 

### Key Messages

In our small single-site, randomized, double-blind, placebo-controlled clinical trial that enrolled 32 participants, anakinra added to dexamethasone did not significantly impact survival in COVID-19 pneumonia CSS patients;The serum levels of IL-6 were significantly higher in patients who did not survive compared to patients who did survive;The treatment with anakinra was associated with decreases in CXCL9 levels;Additional studies are needed to assess the efficacy, timing, dosing, and duration of anakinra treatment in select COVID-19 CSS patients.

## Figures and Tables

**Figure 1 viruses-15-02036-f001:**
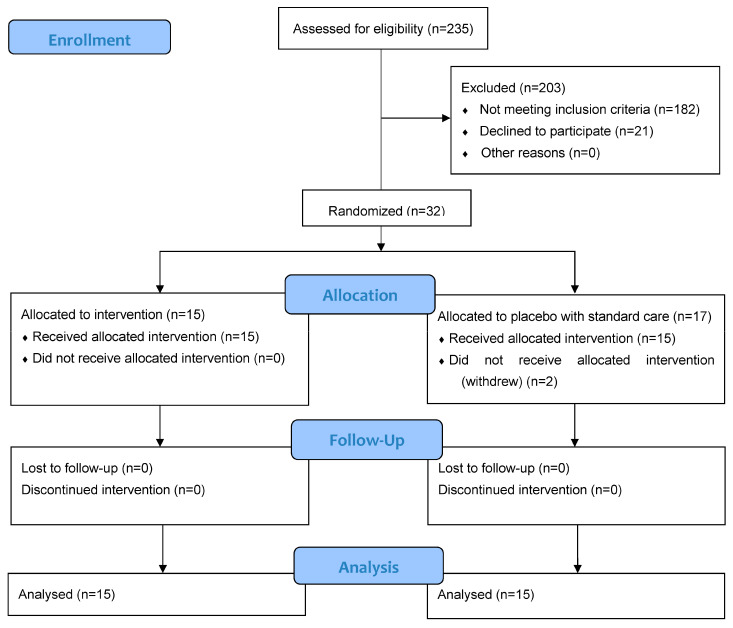
CONSORT 2010 flow diagram showing patient enrollment, randomization, and follow-up.

**Figure 2 viruses-15-02036-f002:**
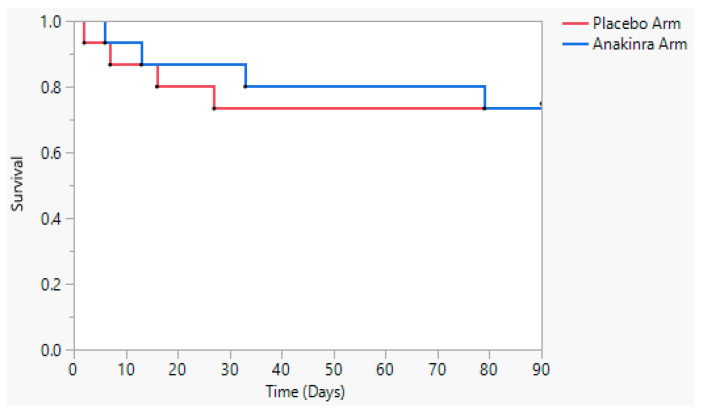
Kaplan–Meier curve demonstrating survival according to treatment in the anakinra and placebo groups. Survival of all individuals exposed to anakinra is shown in blue, and survival of the placebo group is shown in red.

**Table 1 viruses-15-02036-t001:** Baseline demographic characteristics of the participants enrolled in the study and included in the intent-to-treat analysis.

	Anakinra Group (N = 15)	Placebo Group (N = 15)	*p*-Value
**Age, years, mean (SD)**	64.2 (8.2)	61.8 (12.2)	0.5
**Sex, female, N (%)**	5 (33.3)	5 (33.3)	1.0
**Race, N (%)**			
White	11 (73.3)	9 (60.0)	0.7
Black	4 (26.7)	5 (33.3)	
Declined/missing	0 (0)	1 (6.7)	
**BMI, mean (SD)**	29.9 (6.8)	30.5 (3.5)	0.8
**Co-morbidities, N (%)**			
Cardiovascular disease	4 (26.7)	3 (20.0)	1.0
Diabetes mellitus	5 (33.3)	3 (20.0)	0.7
Hypertension	11 (73.3)	10 (66.7)	1.0
Malignancy	1 (6.7)	1 (6.7)	1.0
**Current or former smoker, N (%)**	6 (40.0)	5 (33.3)	1.0

SD, standard deviation.

**Table 2 viruses-15-02036-t002:** Baseline clinical characteristics of the participants enrolled in the study and included in the intent-to-treat analysis.

	Anakinra Group (N = 15)	Placebo Group (N = 15)	*p*-Value
**Clinical inclusion parameters at enrollment, N (%)**			
Elevated D-dimer (>500 ng/mL)	10 (66.7)	11 (73.3)	1.0
Thrombocytopenia (<130 × 10^3^/mm^3^)	2 (13.3)	1 (6.7)	1.0
Leukopenia (WBC < 3.5 × 10^3^/mm^3^) or lymphopenia (<1 × 10^3^/mm^3^)	10 (66.7)	13 (86.7)	0.4
Elevated AST or ALT (>2X ULN)	5 (33.3)	2 (13.3)	0.4
Elevated LDH (>2X ULN)	7 (46.7)	3 (20.0)	0.2
CRP > 100 mg/L	10 (66.7)	11 (73.3)	1.0
**Other medications at enrollment, N (%)**			
Dexamethasone	15 (100.0)	15 (100.0)	N/A
Azithromycin	10 (66.67)	8 (53.33)	0.7
Remdesivir	12 (80.0)	14 (93.3)	0.6
β-lactams	13 (86.7)	10 (66.7)	0.4
Therapeutic anticoagulation at baseline	3 (20.0)	0 (0.0)	0.2
Prophylactic anticoagulation at enrollment	12 (80.0)	15 (100.00)	0.2
**Pulmonary compromise at enrollment ***	9 (60.0)	7 (46.7)	0.7
**MICU patient at enrollment**	4 (26.7)	4 (26.7)	1.0
**Laboratory values at enrollment, mean (SD)**			
CRP, mg/L (normal < 10.9)	130.1 (62.3)	180.8 (113.0)	0.1
Ferritin, ng/mL (normal < 336.2)	1344.1 (711.2)	1922.4 (2304.3)	0.4
LDH, IU/L (normal 120–240)	480.7 (211.3)	414.1 (152.5)	0.3
D-dimer, ng/mL	1262 (1863.9)	2080.7 (3712.2)	0.5
WBC, ×10^3^/mm^3^ (normal 4–11)	8.2 (3.7)	11.1 (4.3)	0.05
Lymphocyte count, ×10^3^/mm^3^ (normal 1.25–5.8)	0.83 (0.4)	0.65 (0.3)	0.2
Platelet count, ×10^3^/mm^3^ (normal 150–400)	273.3 (107.8)	272.5 (99.1)	1.0
Fibrinogen, mg/dL	635.9 (152.1)	636.9 (178.2)	1.0
INR	1.2 (0.2)	1.1 (0.1)	0.2
Triglyceride level, mg/dL (normal 40–150)	131.3 (46.4)	165.6 (79.0)	0.2
AST, IU/L (normal 12–39)	54 (23.4)	44.3 (41.2)	0.4
ALT, IU/L (normal 7–52)	71.6 (54.6)	54.0 (73.4)	0.5
Creatinine, mg/dL (normal 0.7–1.3)	1.1 (0.9)	1.0 (0.5)	0.7

* Defined as patient requiring oxygen support >6 L per minute via nasal cannula. ALT, alanine transaminase; AST, aspartate transaminase; CRP, C-reactive protein; INR, international normalized ratio; IU, international unit; LDH, lactate dehydrogenase; MICU, medical intensive care unit; N/A, not applicable; ULN, upper limit of normal; WBC, white blood cell count.

**Table 3 viruses-15-02036-t003:** Patient outcomes in the anakinra and placebo groups, intent-to-treat (ITT) analysis; N (%) represented unless otherwise stated.

	Anakinra Group (N = 15)	Placebo Group (N = 15)	*p*-Value
**Primary Outcomes**			
Survived hospitalization	11 (73.3)	11 (73.3)	1.0
Did not require mechanical ventilation	10 (66.7)	10 (66.7)	1.0
**Secondary Outcomes**			
Did not require escalation of oxygen support after Day 2, N = 30	8 (53.3)	9 (60.0)	0.7
25% improvement in CRP and ferritin by Day 2, N = 29	6 (40.0)	8 (57.1)	0.4
75% improvement from Day 0 CRP and ferritin elevations by Day 10, N = 26 *	6 (46.15)	7 (53.9)	0.7
75% improvement from Day 0 LDH and D-dimer elevations by Day 10, N = 26 *	0 (0.0)	7 (53.9)	**0.005**
Requiring no more than 2 L/min NC of oxygen support to maintain oxygen saturation ≥93% by Day 10, N = 27	8 (57.1)	9 (69.2)	0.7
Able to maintain oxygen saturation ≥93% on RA by Day 10, N = 27	4 (28.6)	9 (69.2)	0.06
Culture-positive infections, any	4 (26.7)	2 (13.3)	0.7
Bacteremia/fungemia	1 (6.7)	0 (0.0)	1.0
Colonization	3 (20.0)	2 (13.3)	1.0

Categorical variables are reported as absolute number (percentage). *p* values < 0.05 are indicated in boldface. * Missing data for 1 participant. CRP, C-reactive protein; LDH, lactate dehydrogenase; NC, nasal cannula; RA, room air.

**Table 4 viruses-15-02036-t004:** Cytokine levels in the anakinra and placebo groups at Day 0 and Day 10. Mean with standard error of the mean reported, unless otherwise specified.

		Day 0 Levels			Day 10 Levels *		
Cytokine	Reference ULN	Anakinra Group	Placebo Group	*p*-Value	Anakinra Group	Placebo Group	*p*-Value
sCD163	<1785	1384.8 ± 181.8	1382.9 ± 181.2	1.0	1464.5 ± 209.9	1285.0 ± 176.2	0.5
sCD25	<858	1295.7 ± 192.1	1465.9 ± 168.0	0.5	1578.9 ± 517.4	916.5 ± 113.8	0.2
IL-12	<1.9	2.0 ± 0.1	1.9 ± 0.0	0.3	4.0 ± 2.1	1.90 ± 0.0	0.3
IFN-γ	<4.2	4.2 ± 0.0	4.2 ± 0.0	1.0	4.2 ± 0.0	4.2 ± 0.0	0.3
IL-4	<2.2	2.2 ± 0.0	2.2 ± 0.0	1.0	2.2 ± 0.0	2.2 ± 0	1.0
IL-5	<2.1	2.3 ±0.2	2.1 ± 0	0.2	3.7 ± 1.4	2.2 ± 0.1	0.3
IL-10	<2.8	17.4 ± 3.9	9.5 ± 1.2	0.07	15.9 ± 7.2	9.9 ± 5.0	0.5
IL-13	<2.3	2.5 ± 0.7	3.0 ± 0.9	0.7	3.8 ± 1.4	4.2 ± 1.6	0.9
IL-1β	<6.7	7.1 ± 0.6	6.5 ± 0.0	0.3	6.6 ± 0.1	7.2 ± 0.7	0.4
IL-6	<2.0	5.3 ± 0.9	12.6 ± 6.7	0.3	14.6 ± 6.8	16.2 ± 11.0	0.9
IL-8	<30	3.0 ± 0.0	3.0 ± 0.0	1.0	4.6 ± 1.6	3.6 ± 0.6	0.5
TNF	<7.2	3.6 ±0.6	3.0 ± 0.8	0.5	8.3 ± 5.0	2.2 ± 0.3	0.3
IL-2	<2.1	2.1 ± 0.0	2.1± 0.0	1.0	2.6 ± 0.5	2.10 ± 0.0	0.3
IL-17	<1.4	1.4 ± 0.0	1.4 ± 0.0	0.8	5.9 ± 3.1	1.4 ± 0.0	0.2
IL-18	<540	1011.9 ± 121.1	825.3 ± 139.3	0.3	Unavailable	Unavailable	-
CXCL9	<121	465.9 ± 100.1	200.0 ± 42.0	**0.03**	169.0 ± 29.6	143.8 ± 27.3	0.6

TNF, tumor necrosis factor; ULN, upper limit of normal. * Data missing for 3 participants due to death prior to Day 10. *p* values < 0.05 are indicated in boldface.

**Table 5 viruses-15-02036-t005:** Adverse and serious adverse events; N (%) represented unless otherwise stated.

	Anakinra Group (N = 15)	Placebo Group (N = 15)	*p*-Value
**Serious adverse events**			
Acute respiratory distress syndrome	9 (60.0)	7 (46.7)	0.5
Arterial ischemia	4 (26.7)	5 (33.3)	0.7
Sudden cardio-respiratory arrest	4 (26.7)	4 (26.7)	1.0
**Other Adverse Events of Interest**			
Pulmonary embolism or deep venous thrombosis	4 (26.7)	3 (20.0)	0.7
Acute renal failure	7 (46.7)	6 (40.0)	0.7
Neutropenia	1 (6.7)	0 (0.0)	0.3
Anemia	14 (93.3)	13 (86.7)	0.5
Thrombocytopenia	5 (33.3)	5 (33.3)	1.0
Gastrointestinal bleeding	1 (6.7)	0 (0.0)	0.3

## Data Availability

The data underlying this article cannot be shared publicly for the privacy of individuals that participated in the study. The de-identified data will be shared on reasonable request to the corresponding author.

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
