# Peer review of "Prediction of Survival by IL-6 in a Randomized Placebo-Controlled Trial of Anakinra in COVID-19 Cytokine Storm"

_viruses, 2023, doi:10.3390/v15102036_

Round 1

Reviewer 1 Report

In this paper, the authors hav analyse on a cohort of 32 patients, the interest of anakinra on the mortality and the need for intubation or mechanical ventilation in patients with severe COVID-19. They conclude that there was no effect of anakinra on these parameters. 

The main force of this study is that is is a randomized trial. The main weakness of this study, as discussed by the authors, is the number of patients included. Patient recruitment took place from August 2020, a period from which there was good knowledge of the percentage of deaths with severe forms of COVID-19, and the percentage of patients requiring ventilation/intubation. Therefore, it was possible to calculate the number of patients to include to see a treatment effect (for example a 20% reduction in mortality). It seems important to have a more rigourous approach of  the statistics, and to discuss the feasibility of such a study in terms of patients.

Minor comment:

- the number of patients recruted should be indicated in the "key message" section.

Author Response

Reviewer #1:

In this paper, the authors hav analyse on a cohort of 32 patients, the interest of anakinra on the mortality and the need for intubation or mechanical ventilation in patients with severe COVID-19. They conclude that there was no effect of anakinra on these parameters. 

The main force of this study is that is is a randomized trial. The main weakness of this study, as discussed by the authors, is the number of patients included. Patient recruitment took place from August 2020, a period from which there was good knowledge of the percentage of deaths with severe forms of COVID-19, and the percentage of patients requiring ventilation/intubation. Therefore, it was possible to calculate the number of patients to include to see a treatment effect (for example a 20% reduction in mortality). It seems important to have a more rigourous approach of  the statistics, and to discuss the feasibility of such a study in terms of patients.

 Authors Response:  We thank the reviewer for their feedback. While the study commenced enrollment in August of 2020, the study was conceptualized and funding applied for in April of 2020, at which time the U.S. mortality data was not yet well established. We performed an initial power and sample size calculation based upon the initial mortality assessments relevant in the Spring of 2020 (before steroids were standard of care for hospitalized subjects). We have adjusted the statistical analysis section of the methods on pages 3 and 4 which now reads: “We assumed the subset of admitted patients with CSS attributes would have a very high mortality (80%), and wishing to determine if anakinra use added to standard of care would be significantly impactful, we determined a sample size of 32 participants (16 in each group) would provide adequate power of 80% using a two-side alpha of 0.05 to detect a clinically relevant reduction in mortality of 60% between the two groups.”

In addition, we have added the following to the limitations section on page 12: “Appropriate sample sizes could not be accurately estimated when the trial was being planned near the onset of the COVID-19 pandemic. However, at the time of study conceptualization, the in-hospital mortality was estimated to be approximately 80% among those admitted with CSS features. With the advent of routine dexamethasone use between the time of study conceptualization, submission for regulatory approvals, and the start of study enrollment, mortality was not as high as originally projected and the study as designed became underpowered.”

Minor comment:

- the number of patients recruted should be indicated in the "key message" section.

Authors Response:  We have added this to the “key message” section as suggested on page 13, which now reads: “In our small single site randomized, double-blind, placebo-controlled clinical trial that enrolled 32 participants, anakinra added to dexamethasone did not significantly impact survival in COVID-19 pneumonia CSS patients”

Reviewer 2 Report

I applaud the authors for attempting to look at a true placebo controlled randomized clinical trial attempting to assess the utility of using anakinra as an adjuvant to treatment of a very specific group of COVID-19 patients, those meeting criteria for hyper inflammatory cytokine storm syndrome. 

Pros - truly randomized and controlled, patients received more or less standardized therapy especially with regard to the anakinra dosing. The patients were well matched, may have been slightly more severe in the COVID-19 group based on increase in diabetes, LDH and few other factors. 

You show the ability of anakinra to decrease the level of CXCL9 which is significant but is only one of many inflammatory markers, most which did not appear to respond to therapy. It may be possible that the inflammatory cascade, which is driven by many factors is too progressed for anakinra to make a difference at the advanced stage that your patients were enrolled as noted by the elevated IL-6 levels in patients who died.  

Limitations - Their case definition is extremely narrow, therefore, I believe their conclusions can only be towards that limited group with CSS. Comparing to the results of other studies where the entry criteria is much different may indicate that you are treating very different stages of the disease. 

The authors reference how the results differ from other studies to include a large systematic review.  There are major differences that should be noted in the discussion to include that the systematic review showed "The mortality benefit was similar across subgroups regardless of comorbidities (ie, diabetes), ferritin concentrations, or the baseline PaO2/FiO2. In a subgroup analysis, anakinra was more effective in lowering mortality in patients with CRP concentrations higher than 100 mg/L (OR 0·28 [95% CI 0·17-0·47]). Anakinra showed a significant survival benefit when given without dexamethasone (OR 0·23 [95% CI 0·12-0·43]), but not with dexamethasone co-administration (0·72 [95% CI 0·37-1·41])." Since all of your patients received dexamethasone - you should at least note that it could be playing a role. Additionally, the standard deviations on your patients CRP clearly place some of them with a CRP much less than 100 that the systematic review notes is the level needed to see improvement. 

Similarly, treating severe COVID-19 is very different on a month to month or institution to institution basis.  I think you did a great job controlling your variables but I find it difficult to believe that tocilizumab or sarilumab or other anti-inflammatories were not used. Can you comment on the use of other inflammatory mediators and treatments as you did with remdesivir?

You note your small sample size perhaps playing a role in findings that you had but in your design or statistical analysis sections, you do not comment on what enrollment you had desired to achieve, is your study underpowered to show accurately comment on your objectives.

Line 310 - that should be than

Author Response

Reviewer #2:

I applaud the authors for attempting to look at a true placebo controlled randomized clinical trial attempting to assess the utility of using anakinra as an adjuvant to treatment of a very specific group of COVID-19 patients, those meeting criteria for hyper inflammatory cytokine storm syndrome. 

Pros - truly randomized and controlled, patients received more or less standardized therapy especially with regard to the anakinra dosing. The patients were well matched, may have been slightly more severe in the COVID-19 group based on increase in diabetes, LDH and few other factors. 

You show the ability of anakinra to decrease the level of CXCL9 which is significant but is only one of many inflammatory markers, most which did not appear to respond to therapy. It may be possible that the inflammatory cascade, which is driven by many factors is too progressed for anakinra to make a difference at the advanced stage that your patients were enrolled as noted by the elevated IL-6 levels in patients who died.  

Limitations - Their case definition is extremely narrow, therefore, I believe their conclusions can only be towards that limited group with CSS. Comparing to the results of other studies where the entry criteria is much different may indicate that you are treating very different stages of the disease. 

The authors reference how the results differ from other studies to include a large systematic review.  There are major differences that should be noted in the discussion to include that the systematic review showed "The mortality benefit was similar across subgroups regardless of comorbidities (ie, diabetes), ferritin concentrations, or the baseline PaO2/FiO2. In a subgroup analysis, anakinra was more effective in lowering mortality in patients with CRP concentrations higher than 100 mg/L (OR 0·28 [95% CI 0·17-0·47]). Anakinra showed a significant survival benefit when given without dexamethasone (OR 0·23 [95% CI 0·12-0·43]), but not with dexamethasone co-administration (0·72 [95% CI 0·37-1·41])." Since all of your patients received dexamethasone - you should at least note that it could be playing a role. Additionally, the standard deviations on your patients CRP clearly place some of them with a CRP much less than 100 that the systematic review notes is the level needed to see improvement. 

Authors Response: Thank you for these comments. We have added the following text to the discussion on page 11 to address these comments:  “Of note, this review concluded that anakinra was associated with survival benefit primarily when given without concomitant dexamethasone. All patients in our study received dexamethasone, as this became standard of care during the study duration, and thus this likely played a role in improving outcomes in our population. Moreover, the authors found that anakinra was particularly effective in decreasing risk of mortality in people with CRP concentrations higher than 100 mg/L. However, approximately 30% of our population had enrollment CRP concentrations less than 100 mg/L (the level that was associated with mortality benefit in the systematic review), a finding which may help explain the lack of significant benefit seen in our study.”

Similarly, treating severe COVID-19 is very different on a month to month or institution to institution basis.  I think you did a great job controlling your variables but I find it difficult to believe that tocilizumab or sarilumab or other anti-inflammatories were not used. Can you comment on the use of other inflammatory mediators and treatments as you did with remdesivir?

Authors Response:  At the time this trial was conducted very early in the COVID-19 pandemic, it was unclear whether IL-6 blockade would be associated with improved outcomes and was not a part of standard of care. In addition, since this was a clinical trial, we intentionally excluded patients receiving any other immunomodulatory agents. We have added the following text to the methods on page 2 to address this point:  “Use of immunosuppressants such as tocilizumab or sarilumab were not standard of care at the time this trial was conducted very early in the COVID-19 pandemic, and we excluded patients receiving any other immunomodulatory agents.”

You note your small sample size perhaps playing a role in findings that you had but in your design or statistical analysis sections, you do not comment on what enrollment you had desired to achieve, is your study underpowered to show accurately comment on your objectives.

Authors Response:  We have edited the methods section as suggested. The statistical analysis section of the methods now reads: “We assumed the subset of admitted patients with CSS attributes would have a very high mortality (80%), and wishing to determine if anakinra use added to standard of care would be significantly impactful, we determined a sample size of 32 participants (16 in each group) would provide adequate power of 80% using a two-side alpha of 0.05 to detect a clinically relevant reduction in mortality of 60% between the two groups.”

Comments on the Quality of English Language:  Line 310 - that should be than

Authors Response:  We have edited this as suggested. This sentence on page 12 now reads: “In two open-label randomized controlled trials of anakinra in COVID-19 associated CSS, the dose and frequency of anakinra administered was lower than the protocol used in this current study [21, 29].”

Reviewer 3 Report

In this study conducted by Jackson, L.E. et al., the authors assess whether anakinra treatment, along with standard of care treatment, can improve the outcomes of patients with COVID-19 following a highly defined set of criteria. While I'm sure it is unfortunate to see that the addition of anakinra treatment did not reach the endpoints of the study, the reporting done by the authors is extremely detailed and very well done. The paper is extremely well-written. From start to finish, the paper is easy to follow and the methods applied are logical and scientific. For this reviewer, one of the major advantages to this study is how well the placebo group and treatment group have been matched. Although the sample size for this study is relatively small, the attention to detail with group matching and follow up gives the paper a lot of power to support the conclusions drawn. I believe this paper adds important value to the consideration of continuing to use anakinra treatment in patients with COVID-19. I would like to congratulate the authors on a very well-written paper. The only minor change I would make is to expand COVID-19 in the introduction to "coronavirus disease 2019 (COVID-19)" as COVID-19 is an abbreviation. 

Author Response

Reviewer #3:

In this study conducted by Jackson, L.E. et al., the authors assess whether anakinra treatment, along with standard of care treatment, can improve the outcomes of patients with COVID-19 following a highly defined set of criteria. While I'm sure it is unfortunate to see that the addition of anakinra treatment did not reach the endpoints of the study, the reporting done by the authors is extremely detailed and very well done. The paper is extremely well-written. From start to finish, the paper is easy to follow and the methods applied are logical and scientific. For this reviewer, one of the major advantages to this study is how well the placebo group and treatment group have been matched. Although the sample size for this study is relatively small, the attention to detail with group matching and follow up gives the paper a lot of power to support the conclusions drawn. I believe this paper adds important value to the consideration of continuing to use anakinra treatment in patients with COVID-19. I would like to congratulate the authors on a very well-written paper. The only minor change I would make is to expand COVID-19 in the introduction to "coronavirus disease 2019 (COVID-19)" as COVID-19 is an abbreviation. 

Authors Response:  We thank the reviewer for their very positive feedback. As suggested, we have adjusted the first sentence of the introduction on page 1, which now reads:  “The respiratory illness coronavirus disease 2019 (COVID-19) caused by severe acute respiratory syndrome coronavirus 2 (SARS-CoV-2) induces moderate to severe disease requiring hospital admission and supplemental oxygen support in approximately 10-15% of patients [1-3].”